# Dynamic electron correlations with charge order wavelength along all directions in the copper oxide plane

F. Boschini [1,2,3], M. Minola [4], R. Sutarto [5], E. Schierle [6], M. Bluschke[4,6], S. Das[7], Y. Yang[7], M. Michiardi[1,2,8], Y. C. Shao[9], X. Feng[9], S. Ono[10], R. D. Zhong[11], J. A. Schneeloch[11], G. D. Gu[11], E. Weschke[6], F. He [5], Y. D. Chuang[9], B. Keimer [4], A. Damascelli [1,2], A. Frano [7] & E. H. da Silva Neto [12,13,14✉]

In strongly correlated systems the strength of Coulomb interactions between electrons, relative to their kinetic energy, plays a central role in determining their emergent quantum mechanical phases. We perform resonant x-ray scattering on $Bi_2Sr_2CaCu_2O_{8+\delta}$, a prototypical cuprate superconductor, to probe electronic correlations within the $CuO_2$ plane. We discover a dynamic quasi-circular pattern in the $x$-$y$ scattering plane with a radius that matches the wave vector magnitude of the well-known static charge order. Along with doping- and temperature-dependent measurements, our experiments reveal a picture of charge order competing with superconductivity where short-range domains along $x$ and $y$ can dynamically rotate into any other in-plane direction. This quasi-circular spectrum, a hallmark of Brazovskii-type fluctuations, has immediate consequences to our understanding of rotational and translational symmetry breaking in the cuprates. We discuss how the combination of short- and long-range Coulomb interactions results in an effective non-monotonic potential that may determine the quasi-circular pattern.

[1] Quantum Matter Institute, University of British Columbia, Vancouver, BC V6T 1Z4, Canada. [2] Department of Physics & Astronomy, University of British Columbia, Vancouver, BC V6T 1Z1, Canada. [3] Centre Énergie Matériaux Télécommunications, Institut National de la Recherche Scientifique, Varennes, QC J3X 1S2, Canada. [4] Max Planck Institute for Solid State Research, Heisenbergstrasse 1, D-70569 Stuttgart, Germany. [5] Canadian Light Source, Saskatoon, SK S7N 2V3, Canada. [6] Helmholtz-Zentrum Berlin für Materialien und Energie, BESSY II, Albert-Einstein-Str. 15, 12489 Berlin, Germany. [7] Department of Physics, University of California San Diego, La Jolla, CA 92093, USA. [8] Max Planck Institute for Chemical Physics of Solids, Nöthnitzer Straße 40, Dresden 01187, Germany. [9] Advanced Light Source, Lawrence Berkeley National Laboratory, Berkeley, CA 94720, USA. [10] Central Research Institute of Electric Power Industry, Yokosuka, Kanagawa 240-0196, Japan. [11] Condensed Matter Physics and Materials Science, Brookhaven National Laboratory, Upton, NY, USA. [12] Department of Physics, University of California, Davis, CA 95616, USA. [13] Department of Physics, Yale University, New Haven, CT 06511, USA. [14] Energy Sciences Institute, Yale University, West Haven, CT 06516, USA. ✉email: eduardo.dasilvaneto@yale.edu

The ability to obtain information about the interacting potential from experimentally measured correlation functions is key to advancing our understanding of many-body systems. This is a central challenge to several problems in physics, ranging from investigations of the universe through correlations in the cosmic microwave background, to our understanding of strongly correlated electrons in solids[1–3]. In cuprate high-temperature superconductors, a quintessential solid-state many-body system, it is important to understand the form of the electron–electron interactions and how they lead to various ordered states such as charge order and superconductivity. Many theoretical descriptions of the cuprates focus on different sectors of the Coulomb interaction, separating the onsite, short-range, and long-range components. In undoped compounds, strong onsite Coulomb interactions result in a Mott insulator with an antiferromagnetic ground state. Doping destabilizes this ground state and promotes high-temperature superconductivity. Although recent numerical simulations indicate that fluctuating stripe patterns may emerge from a three-band Hubbard model with only on-site Coulomb repulsion[4], long-range Coulomb interactions have long been proposed to avoid phase segregation between undoped and hole-rich domains[5]. At the same time, Coulomb interactions between $CuO_2$ planes also appear to be essential to explain the recently discovered three-dimensional plasmons in electron-doped cuprates[6,7]. The large range of models proposed to describe various aspects of the cuprates reveal the need for evermore detailed measurements of the electron correlations that can be used to narrow down the Hamiltonian parameter space.

A particular manifestation of spatial correlations among electrons in the cuprates is charge order (CO), a phenomenon where electrons self-organize into spatial patterns with periods ranging from five to three lattice constants[8–16]. Theoretically, CO can emerge as a consequence of the Coulomb interaction experienced by electrons within the $CuO_2$ plane[4,17–21]. In addition, the band structure—which may be affected by the Coulomb repulsion—may also play an important role in determining the CO $\mathbf{q}$ structure. More generally, the spatial and time dependence of the electron correlations encodes information about both Coulomb interactions and the band structure. Thus, as an immediate benefit of recognizing the signatures of Coulomb interactions in measurements of electron correlations, it may also be possible to conclusively resolve the microscopic origin of the CO.

Aiming to better understand the spatial and dynamic behavior of electron correlations in the cuprates, we carried out a series of resonant x-ray scattering (RXS) experiments across a range of temperature and doping values in $Bi_2Sr_2CaCu_2O_{8+\delta}$ (Bi2212). This is the ideal cuprate for our experiments since its band structure parameters and Fermi surface topology are well known from angle-resolved photoemission spectroscopy. Unlike most of the previous x-ray experiments on the cuprates, which have either focused on the high-symmetry directions only ($q_x$, $q_y$ and $q_x = q_y$) or were performed without energy-loss resolution, we fully mapped the $q_x$–$q_y$ plane with inelastic scattering.

Our main result is the identification of a distinct dynamic "ring-like" scattering pattern in the $q_x$–$q_y$ plane. Remarkably, this ring has a wave vector radius that matches the periodicity of the static CO, indicating that the two phenomena are related. We propose that the shape of the dynamic scattering pattern may reflect a simple form of the effective Coulomb potential, $V(\mathbf{q})$, that takes into account both short- and long-range interactions. We also attempt to model the intensity anisotropy of the ring-like feature in the $q_x$–$q_y$ plane by including band structure effects in a random phase approximation (RPA) calculation. However, we find that the RPA fails to capture the shape and intensity anisotropy at a qualitative level. Moreover, we discuss temperature- and doping-dependent experiments that reveal the nuanced effects of the lattice potential, discommensurations and doping to the stabilization of dynamic charge correlations, therefore providing a comprehensive picture of charge order formation in a cuprate.

## Results

**Quasi-circular correlations in the $q_x$–$q_y$ scattering plane**. We used RXS at the Cu-$L_3$ edge (photon energy ≈ 932 eV) in both energy-integrated (EI-RXS) and energy-resolved (resonant inelastic x-ray scattering, RIXS) modes, where the latter resolves the energy loss of the scattered photon while the former cannot distinguish between static and dynamic signals. Note that our use of the term "static" encompasses both short- and long-range order. To map the $q_x$–$q_y$ plane we took momentum cuts along multiple in-plane directions at several azimuthal angles, $\varphi$, relative to $q_x$. Since the majority of the $\mathbf{q}$-dependent RXS signal originates from orbital and charge-transfer excitations at a relatively high-energy loss ($E > 0.9$ eV, HE), the ability to resolve the CO correlations is significantly improved by isolating the low-energy region ($E < 0.9$ eV, LE), which is possible with RIXS. State-of-the-art RIXS instruments now reach energy resolutions better than 40 meV at the Cu-$L_3$ edge[22], but only at the cost of a lower scattering throughput. This actually significantly increases the acquisition time necessary to detect CO correlations in Bi2212 and makes the length of a full $q_x$–$q_y$ mapping prohibitive. The qRIXS instrument at the Advanced Light Source (ALS) provides the necessary compromise, combining high throughput and an energy resolution of $\Delta E = 800$ meV (full-width-at-half-maximum), which is enough to remove the high-energy contribution and improve the signal-to-background ratio of our measurements.

Figure 1a shows the RIXS signal as a function of $q = |\mathbf{q}|$ for some values of $\varphi$ between 0° and 90°, measured at 50 K and separated into the LE and HE signals by integrating the spectral weight (highlighted regions in the top-left inset of Fig. 1a). Noticing that the HE scattering shows only a featureless background, we focus on the LE signal. For $\varphi = 0°$ and 90°, the 50 K data show a clear peak at $\overline{q} = 0.27$ rlu, consistent with previous experiments[14]. Remarkably, a peak is also present for $\varphi = 45°$, with the scattering pattern along this direction being peaked at nearly the same $|\mathbf{q}| = \overline{q}$. Indeed, the full $q_x$–$q_y$ structure, Fig. 1b, reveals a ring-like scattering feature, with stronger intensity along $\varphi = 0°$. To correct for geometrical and systematic effects, such as variations in the detector efficiency throughout the measurements, we normalize the data by the total fluorescence collected at the detector (i.e., the energy-integrated RIXS spectrum at each angle, Supplementary Note 2). This yields the $q_x$–$q_y$ map in Fig. 1c which more clearly reveals the quasi-circular in-plane structure—note that given the width of the scattering feature along with $|\mathbf{q}|$ in the integrated LE signal we cannot rule out small deviations from a perfect circle. Nevertheless, these findings clearly establish the presence of a ring-like in-plane scattering pattern with radius $\overline{q}$.

The origin of the ring-like scattering feature may be deeply tied to the underlying functional form of the Coulomb interaction between valence electrons, $V(\mathbf{q})$—the absence of the ring-like feature in the HE signal indicates that interactions with the core-hole potential are not responsible for the quasi-circular scattering pattern (Supplementary Note 1). While we cannot directly resolve the full form of the interacting potential in $\mathbf{q}$ space, we propose that a $V(\mathbf{q})$ with minima that form a quasi-circular scattering pattern in the $q_x$–$q_y$ plane may be the simplest possible explanation for our observations. To illustrate this scenario, we consider a $V(\mathbf{q})$ with two contributions: $U(\mathbf{q})$, the short-range residual Coulomb repulsion, and $V_c(\mathbf{q})$, which is the long-range

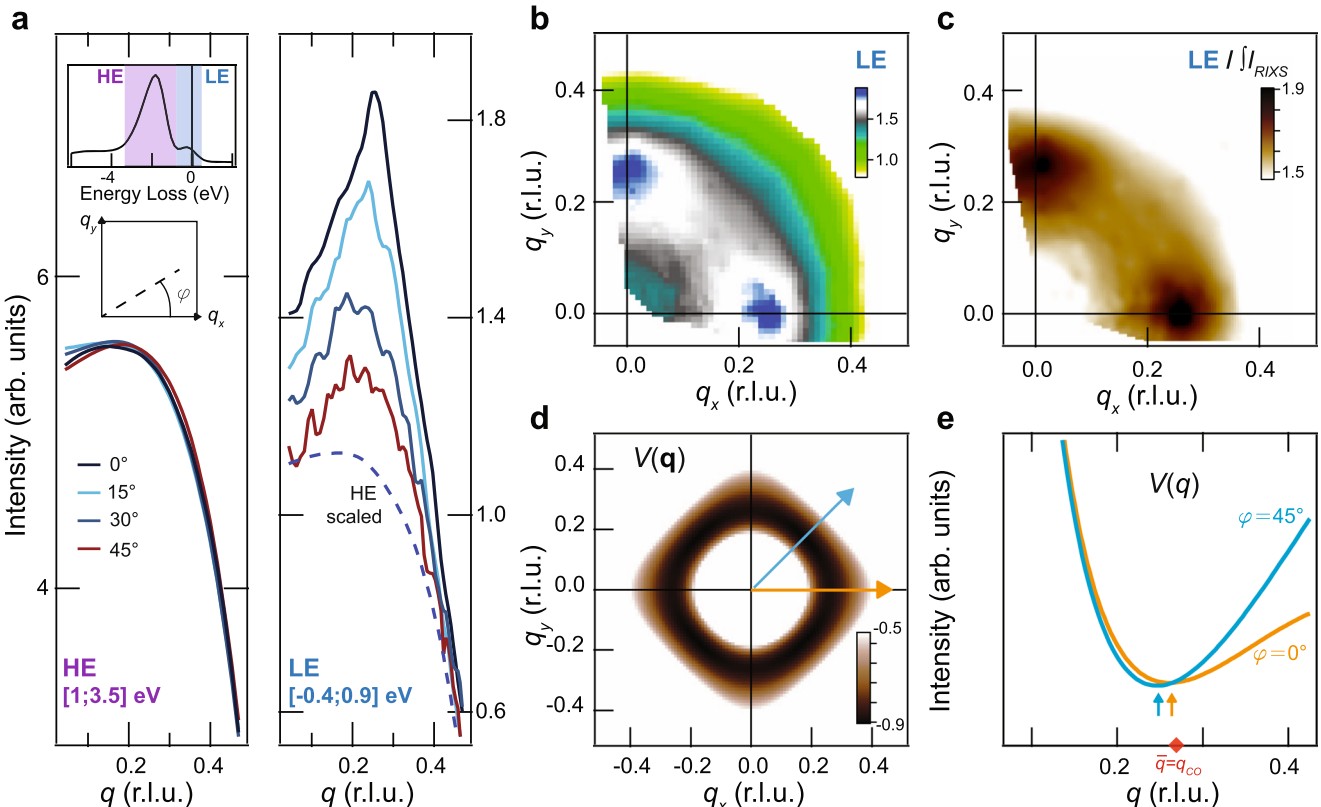

**Fig. 1 The natural tendency for Coulomb interactions to cause a quasi-circular in-plane scattering pattern. a** RIXS scattering measured at 50 K as a function of momentum along different $\varphi$, obtained by integrating the energy loss spectra over the high-energy ($E > 0.9$ eV, HE) and low-energy ($E < 0.9$ eV, LE) regions. The inset displays the Cu-L$_3$ RIXS energy-loss spectrum for $q \approx 0.05$ rlu and $\varphi = 0°$. The blue dashed line compares the featureless and $\varphi$-independent HE scattering signal to the LE curves. **b** CO structure in the $q_x$-$q_y$ plane integrated over the LE region for 50 K. **c** Similar to (**b**) with LE data normalized to the total fluorescence, i.e., the RIXS spectrum integrated in the $[-4,25]$ eV energy range ($\int I_{RIXS}$, details in Supplementary Note 2). **d** The structure of the Coulomb interaction $V(\mathbf{q})$ calculated using known parameters for Bi2212 (details in Supplementary Note 4). **e** Line cuts of (**d**) along $\varphi = 0°$ and 45°, as indicated by the blue and orange arrows. The red diamond in **e** indicates the experimentally determined $\bar{q} = q_{CO}$.

Coulomb interaction projected onto the CuO$_2$ plane[23]. Individually, the two terms have a monotonic **q**-dependence, but together they yield a minimum in the potential near **q**-vectors where they intersect. In turn, this minimum in the effective Coulomb interaction would result in electron–electron correlations at finite **q**, which would be reflected in our RIXS measurements. To visualize the proposed $V(\mathbf{q})$, we perform a proof-of-principle calculation, where the form for the long-range potential is obtained from the solution to Poisson's equation for a 2D plane embedded in a 3D tetragonal lattice. The result in Fig. 1d shows a fourfold symmetric in-plane structure whose minima nevertheless form a quasi-circular contour. In particular, the momentum cuts along with the two high-symmetry in-plane directions, $\varphi = 0°$ and 45°, show that the minima of $V(\mathbf{q})$ occur near the experimentally determined $\bar{q}$, (Fig. 1e). Note that the small deviation of the minima of the calculated $V(\mathbf{q})$ from a quasi-circular contour is still consistent with the experimental data, given the large width of the scattering peaks along all $\varphi$. Importantly, the observed ring structure is incompatible with a Fermi surface origin, which would cause pronounced square-like **q** patterns, with peaks at very different values of $|\mathbf{q}|$ between $\varphi = 0°$ and 45° (see Supplementary Note 5 for a calculation using known band structure parameters[24]). In other words, the form of the Coulomb repulsion described above captures the most novel aspect of our measurements—a quasi-circular scattering structure—and suggests that the inclusion of long-range Coulomb interactions may be necessary for a complete theoretical description of the cuprates.

**Dynamic nature of the correlations at $\bar{q}$.** While the $V(\mathbf{q})$ proposed may capture the shape of the ring-like feature, it is not sufficient to explain the intensity profile in the $q_x$-$q_y$ plane. A closer look at the LE signal in Fig. 1 reveals a stronger peak along $\varphi = 0°$ and 90° than at $\varphi = 45°$. This signal along $q_x$ and $q_y$ represents directional CO correlations locked along $x$ and $y$. Upon first inspection, note that the directional CO peaks at $[q_x = q_{CO}, 0]$ and $[0, q_y = q_{CO}]$ have the same wave-vector magnitude as the quasi-circular scattering patterns (i.e., $\bar{q} = q_{CO}$), which suggests a common origin. The presence of both directional CO and quasi-circular correlations may appear at odds with scanning tunneling spectroscopy (STS) experiments, which show directional CO only[14]. However, we note that while STS is a slow time-scale probe sensitive to static CO only, the fast x-ray scattering process is also sensitive to dynamic correlations as argued above. Thus, the presence of the ring-like scattering in RXS but not in STS automatically implies its dynamic nature. This is confirmed by further separating the low-energy RIXS spectrum into quasielastic ($-200 < E < 200$ meV) and inelastic ($500 < E < 900$ meV) regions, Fig. 2a–c, which shows that the directional CO peaks dominate the elastic scattering regime, while the quasi-circular scattering pattern becomes detectable and pronounced only in the inelastic regime. Note that these energy windows were chosen to illustrate the trend where the intensity contrast between the CO peaks and the ring-like feature decreases with increasing energy, and higher resolution RIXS is necessary to resolve the energy dependence of this intensity anisotropy. Nevertheless, this trend suggests that the lattice acts as a stabilizing force, locking dynamic

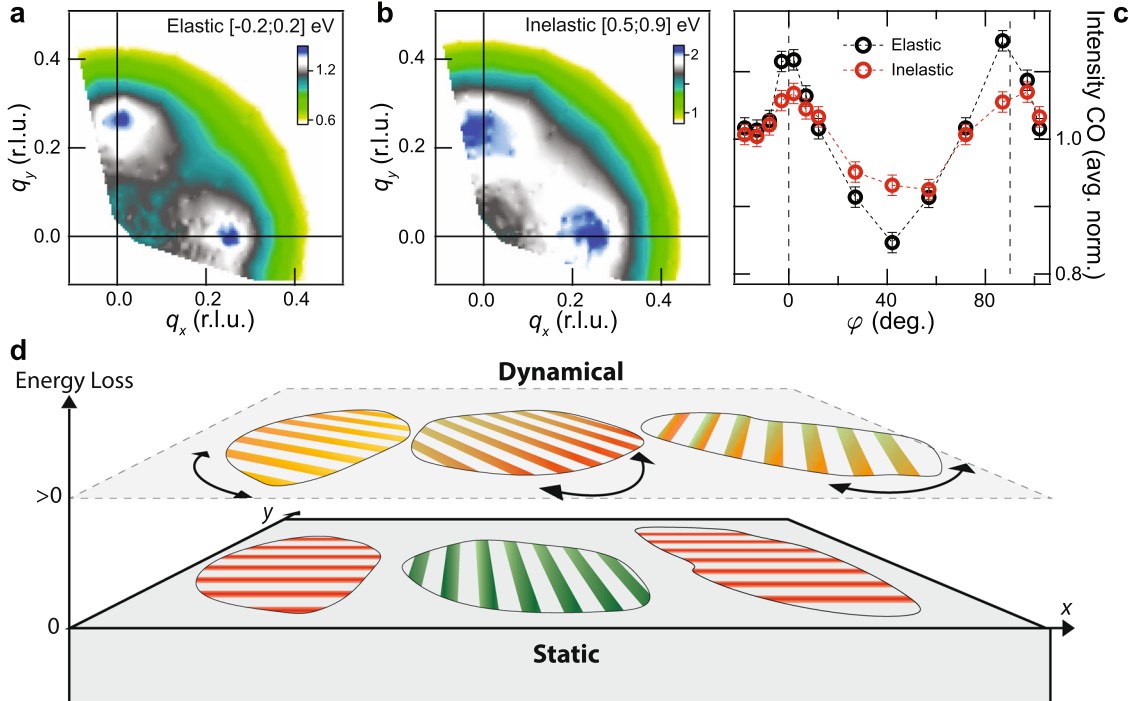

**Fig. 2 The effect of lattice symmetry in locking short-range static charge order. a** Static charge order (CO) structure showing well-defined peaks locked along $q_x$ and $q_y$. **b** Dynamic charge correlations showing the ring-like structure. **c** Scattering intensity at $q = \overline{q} = q_{CO}$ obtained from (**a**) and (**b**), and normalized to their respective averages. The error bars in (**c**) represent the systematic errors associated with the experiment (see Supplementary Note 2). **d** Schematic of CO fluctuations: static short-range CO domains are allowed to dynamically fluctuate into any in-plane direction.

charge correlations from $\overline{q}$ into directional static order along the Cu–O bond direction. In Bi2212, this effect probably occurs at a short length scale, creating alternating stripe domains as reported by STS experiments[25]. This inhomogeneous short-range behavior may explain our observation of the simultaneous occurrence of the dynamic ring and the directional order. Domains of the latter are allowed to fluctuate into other directions, as depicted in Fig. 2d, thus forming the quasi-circular dynamic scattering pattern in the $q_x$–$q_y$ plane.

The dynamic charge order observed in our experiments may also provide complementary insights into recently reported momentum-resolved electron energy-loss spectroscopy (MEELS) experiments on Bi2212[26,27]. In our RIXS experiments, the quasi-circular signal appears to only be clearly observed when the spectrum is integrated over a large energy range ($\approx 0.9$ eV range), Fig. 1a, which suggests that the quasi-circular dynamic CO correlations exist over a wide energy range within the mid-infrared energy scale (MIR, 0.1–1 eV). With much better energy resolution, the MEELS measurements identify a spectral enhancement below 0.5 eV with the lowering of temperature for multiple underdoped Bi2212 samples but not for overdoped samples. As we show below, the CO is also most prominent in the underdoped regime. Therefore, the intersection between RIXS and MEELS experiments lead us to conclude that the dynamic CO correlations occur over a large energy scale, ~0.5 eV, in the MIR region.

Both MEELS and RIXS contain information about the dynamic susceptibility, which can be approximated as $\chi(\mathbf{q}, \omega) = \Pi(\mathbf{q}, \omega)/[1 + V(\mathbf{q})\Pi(\mathbf{q}, \omega)]$, where $\Pi(\mathbf{q}, \omega)$ is the polarizability and $V(\mathbf{q})$ is the Coulomb potential. This is a convenient description where charge order can emerge at $\mathbf{q}$ determined by either the minima of $V(\mathbf{q})$ or by peaks in $\Pi(\mathbf{q}, \omega = 0)$. The RIXS experiments suggest a central role played by $V(\mathbf{q})$ in promoting charge order. However, in principle $\Pi(\mathbf{q}, \omega)$ can and should modify the CO in-plane

structure and intensity pattern. For example, in the commonly used RPA, the polarizability is approximated by the Lindhard function, $\Pi_{\mathrm{Lind}}(\mathbf{q}, \omega)$. Its static form $\Pi_{\mathrm{Lind}}(\mathbf{q}, \omega = 0)$ reflects the density of states at the Fermi level and it can be used to describe charge order caused by Fermi surface instabilities. Although $\chi$ cannot be quantitatively obtained from the RIXS measurement without material-specific modeling[28], we can still inquire whether RPA captures qualitative features such as the intensity anisotropy at $|\mathbf{q}| = \overline{q}$. First we note that a calculation of $\Pi_{\mathrm{Lind}}(\mathbf{q}, \omega = 0)$ yields square patterns in $\chi$ that significantly deviate from a circular in-plane scattering structure (Supplementary Note 5). Recognizing that the quasi-circular pattern is seen more clearly in the data after integration over the LE region (Fig. 1), we also integrated the RPA $\chi(\mathbf{q}, \omega)$ over the MIR scale, which results in a structure that is closer to a circular pattern (Supplementary Note 5). This is because the $\mathbf{q}$ space patterns of $\Pi_{\mathrm{Lind}}(\mathbf{q}, \omega)$ disperse and broaden with energy, which causes them to be weakened by the energy summation. However, even the integrated RPA susceptibility still shows a significant deviation from our RIXS measurements, showing a stronger peak intensity along $\varphi = 45°$ instead of along $\varphi = 0°$. Of course, the ARPES-based band structure parameters are not precise over the unoccupied states, which means that we cannot strictly rule out a pure RPA/Lindhard scenario.

We also consider an alternative phenomenological description with polarizability that is featureless in $\mathbf{q}$ space, as recently reported by MEELS experiments that indicate the breakdown of the RPA/Lindhard picture in Bi2212 over the 0.1–1 eV range[27]. Such a featureless $\Pi(\mathbf{q}, \omega)$ would still not account for the intensity anisotropy as a function of $\varphi$, Fig. 2, but it would at least result in the correct shape (i.e., a quasi-circular scattering pattern). Thus, it will be interesting to test whether theoretical descriptions that have been invoked to explain the MEELS experiments and strange metal behavior (e.g., a marginal Fermi liquid[26,29] or a Sachdev-Ye-Kitaev

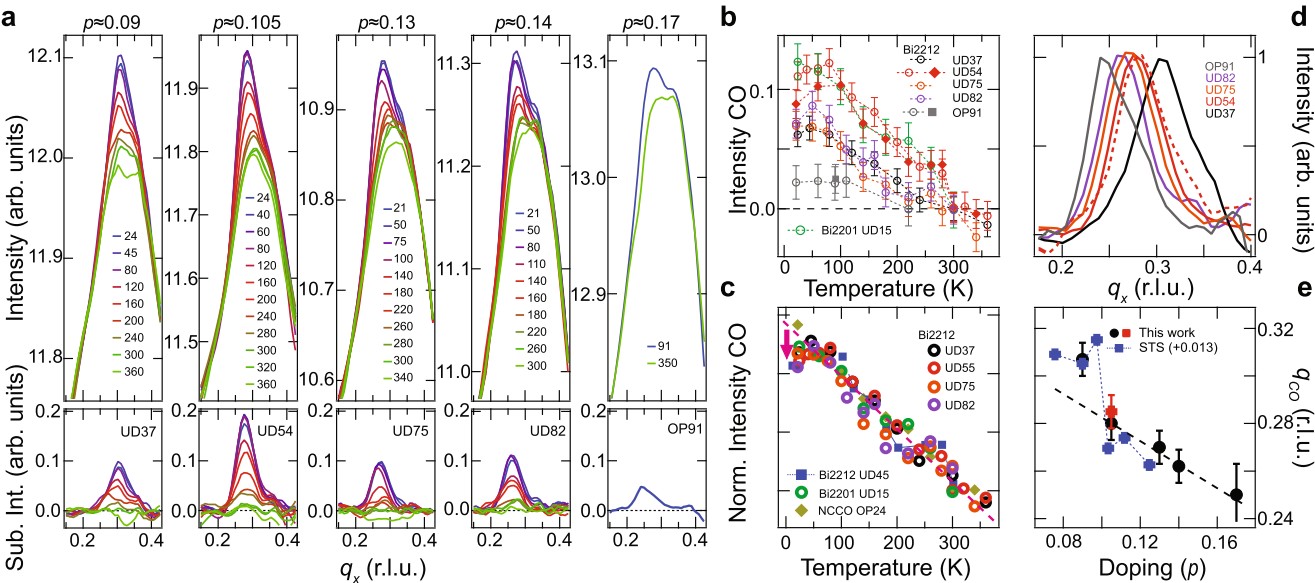

**Fig. 3 Temperature and doping dependence of the directionally locked, short-range static charge order. a** EI-RXS scattering curves for different hole-concentrations and temperatures (top) and the respective charge order (CO) peaks obtained after subtracting the 300 K data (bottom). **b** Temperature dependence of the CO intensity for several dopings of Bi2212 and for underdoped Bi2201. The error bars in (**b**) represent the systematic errors associated with the experiment (see Supplementary Note 2). **c** Temperature dependence of the CO for Bi2212 and Bi2201 at multiple hole doping levels, as well as for the electron-doped cuprate NCCO, scaled to show that the curves collapse on a single line (red dashes) above ~75 K. All samples show a deviation from the linear behavior below 50 K signaling their competition with superconductivity, except for the electron-doped NCCO[16]. The Bi2212 UD45 and NCCO OP24 data are reproduced from refs. [14,59], respectively. **d** Low-temperature CO peaks normalized to their peak value showing the doping dependence of $q_{CO}$. The red dashed and solid curves represent UD54 samples with and without Dy cation substitution, respectively. **e** CO peak location versus hole doping (holes/Cu) obtained from our bulk-sensitive experiments compared to the $q_{CO}$ values obtained from surface-sensitive STS experiments reported in ref. [14]. The red and black points for $p = 0.105$ represent samples with and without Dy cation substitution, respectively. The dashed line in **e** is a guide to the eye. The labels indicate under (UD) and optimally doped (OP) followed by the value of their superconducting transition temperature. The error bars in (**e**) represent the 99% confidence interval of $q_{CO}$ values retrieved by fitting with a Gaussian function.

Hamiltonian with random Coulomb interactions[30,31]) could also explain the dynamic quasi-circular scattering in RIXS.

**Temperature-doping dependence of CO correlations from EI-RXS.** Dynamic CO correlations are expected to play a determining role in forming a dome-shaped phase diagram of static CO[22,32], which has been observed in the bulk of several cuprate families[8]. The dome has not been reported for Bi-based cuprates, although surface-sensitive STS studies show a maximum of the static CO intensity near $p \approx 0.12$ in Bi2212[33]. Figure 3a shows our doping and temperature-dependent EI-RXS measurements. In principle, the putative CO dome could be resolved by an accurate determination of the static CO onset temperature. However, since the RXS signal at $q_{CO}$ comprises both static and dynamic correlations, it is difficult to determine the CO onset temperature from EI-RXS measurements, which show a linear temperature dependence without distinct features above 50 K, Fig. 3b, c. Nonetheless, static charge order dominates the RXS cross-section at low-temperatures along $q_x$ or $q_y$ and is mostly suppressed at room temperature. After subtraction of the high-temperature data (Fig. 3a, bottom), the EI-RXS measurements reveal that the strongest low-temperature CO peak occurs for $p = 0.105$ and is suppressed with doping in either direction.

Our EI-RXS measurements also show that $q_{CO}$ decreases linearly with increasing $p$ for $p > 0.105$, Fig. 3d, e. With the exception of La-based cuprates, this trend is observed in various cuprate works[8], which originally suggested a strong link between the Fermi surface and $q_{CO}$. Instead, our observation of dynamic correlations at $\bar{q} = q_{CO}$ indicates that the charge order periodicity may actually be determined by $V(\mathbf{q})$. Here the doping dependence may originate from, for example, changing dielectric properties

modifying $V(\mathbf{q})$ or may be due to subtle features of the polarization bubble $\Pi$ that could weakly influence the susceptibility $\chi$ around the value of $q_{CO}$ determined by $V(\mathbf{q})$. Such effects are expected to result in a smooth evolution of $q_{CO}$ with doping, as observed for $p > 0.105$. Interestingly, Fig. 3e shows an abrupt jump in $q_{CO}$ across $p \approx 0.1$, consistent with previous STS experiments[14]. To our knowledge, there are no abrupt changes in the band structure or dielectric properties of Bi2212 that can account for this jump. However, we note that at $p = 0.096$ the average distance between holes is $\lambda = 1/\sqrt{p} = 3.22a$, where $a$ is the in-plane lattice constant. In reciprocal space $Q_h = 2\pi/\lambda = 0.31$ rlu which is the observed $q_{CO}$ for $p < 0.1$. Thus, it seems possible that at $p < 0.1$ the CO period locks to the average hole concentration to minimize the energy cost of discommensurations. Upon further doping, $Q_h$ continually increases and diverges from $q_{CO}$, becoming energetically unfavorable for the two to merge.

The temperature-dependent EI-RXS results also contain information about the relationship between charge order and superconductivity. Figure 3c shows the temperature dependence of the re-scaled $q_{CO}$ peak intensity, consistently revealing an intensity drop below ~50 K. This drop may be related to the tendency of the CO to compete with superconductivity, which has been previously demonstrated for other cuprates by RXS experiments[9,11,12,15] and on the surface of Bi2212 by STS[14]. Further contrasting the two techniques, STS shows a much more pronounced suppression of the static CO than what we observe in the EI-RXS. At this point, it remains unclear if this is a dichotomy between surface and bulk properties, or whether it indicates that superconductivity affects dynamic and static correlations at $q_{CO}$ differently. Only future temperature-dependent RIXS, following

the mold of the experiments presented here, will elucidate the relationship between the dynamic CO and superconductivity. Nonetheless, if the minimum in $V(\mathbf{q})$ is directly responsible for short-range CO and the latter competes with superconductivity, it stands to reason that the particular form of $V(\mathbf{q})$ implied by our studies would also be of key relevance to superconductivity.

## Discussion

Over the last few years, RIXS measurements have revealed dynamic correlations associated with CO in various cuprate families[22,34–38]. However, while their detailed relationship to this work remains an open question to be resolved by future $\varphi$-dependent RIXS measurements, a comparison among them reveals deep connections. For instance, in the La-based system, an inelastic signal at momentum values larger than $q_{CO}$ above the onset temperature of static CO was reported[36]. The signal seems to persist beyond the underdoped regime, which has been suggested as evidence of strong correlations driving the CO[38]. In $YBa_2Cu_3O_{6+\delta}$ (YBCO), where the $q_{CO}$ decreases with doping, there have been recent reports of dynamic CO correlations seen by RIXS along the Cu–O bond direction with characteristic energies below 50 meV[22], which also extend to higher doping levels. Larger CO energy scales have yet to be uncovered in YBCO. However, dynamic CO correlations have been observed in the bulk of the electron-doped cuprate $Nd_{2-x}Ce_xCuO_4$ (NCCO) near-optimal doping at the energy scale of paramagnon excitations ($E \approx 0.25$ eV)[34]. It will be interesting to investigate the in-plane topology of that feature with RIXS. Also, a circular scattering feature has been observed in EI-RXS measurements of very underdoped $T'$-$Nd_2CuO_4$ thin films[39]. While those results have been attributed to low-energy glassy charge modulations that reflect the shape of the Fermi surface, they could also be related to the dynamic scattering feature we observe in Bi2212 with RIXS at higher energies. Note that static STS measurements do not report a ring-like glassy structure in Bi2212. In Hg-cuprates, moreover, dynamic CO signals above 70 K also extend to 200 meV and possibly higher[37]. These higher scales observed in NCCO and Hg-cuprates are not only consistent with our measurements but also represent an energy range more compatible with strong correlation physics.

The above comparison between different cuprates suggests that the quasi-circular correlations may be a universal feature of the cuprates. Although we cannot provide a model that captures all the aspects of our experiments, we showed that the inclusion of both short- and long-range interactions, and the resulting minima in $V(\mathbf{q})$, may be essential to explain the shape of the correlations in the $q_x$–$q_y$ plane. Then it is interesting to ask what is the real space form of a non-monotonic $V(\mathbf{q})$ and how general is it. For a simple system of two electrons separated by $r$, the interaction energy follows a simple $1/r$ form. The monotonic behavior of the $1/r$ potential is also manifest in momentum space, exhibiting a $1/q^2$ structure. In solids, the electron is embedded in a crystalline environment, a polarizable medium that modifies the effective electron–electron interactions, Fig. 4a. This modification will happen more strongly at length scales of a few lattice spacings, weakening the short-range potential due to the polarizability of the nearby atoms. At the same time, the long-range behavior of the Coulomb interaction will remain largely unaffected by the details of the local crystal environment. Thus, in general, it is possible for the effective Coulomb interaction $V(\mathbf{r})$ to develop a non-monotonic spatial dependence, perhaps even a minimum, in the range of a few lattice spacings (Fig. 4b). In turn, this may also lead to a minimum in the structure of $V(\mathbf{q})$ (Fig. 4c). Depending on the details of a specific crystal structure, this could lead to an effective attractive potential that may be conducive to

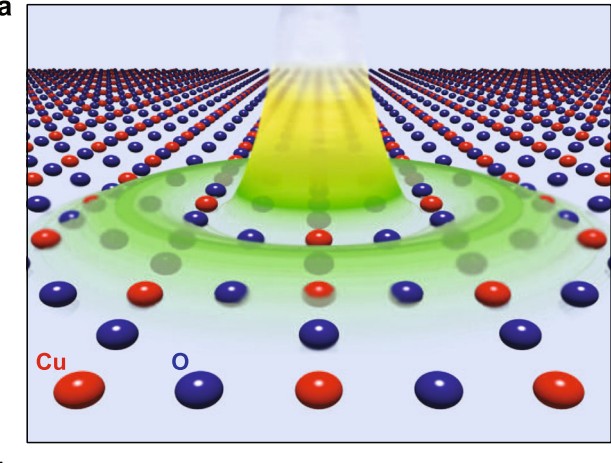

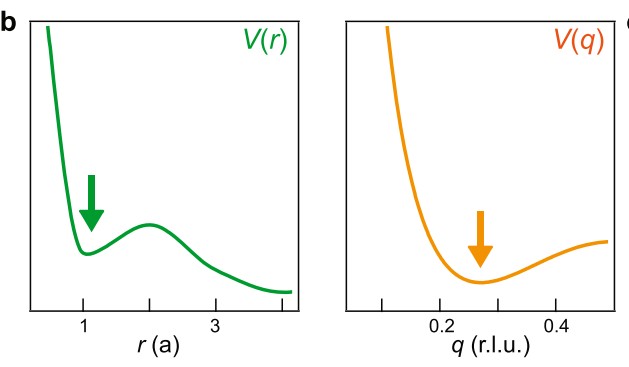

**Fig. 4 Non-monotonic Coulomb interactions. a** Pictorial representation of the Coulomb interaction potential $V(r)$ in the $CuO_2$ plane. The Coulomb potential created by an electron at a Cu site is non-monotonic at the length scale of a few lattice units $a$ (Cu–Cu distance) due to a non-uniform polarizability that weakens the short-range electron–electron interactions. **b**, **c** Schematic profiles of $V(r)$ and $V(q)$ along the Cu–O bond direction. The arrows highlight a minimum at approximately $a$ (green) and $q \approx 0.27$ rlu (orange), respectively.

superconductivity or long-range density wave order[40–44]. Even if the potential is not rendered attractive, the electron density may still self-organize into a periodic structure in order to minimize the Coulomb repulsion, resulting in short-range or fluctuating charge density waves. The effect described above may be broadly applicable to understanding superconductivity and spatial ordering in a variety of systems, including but not restricted to Cu- and Fe-based superconductors[40–44]. General as it may be, the non-monotonic behavior of $V(\mathbf{q})$ has not yet been directly reported in experiments. Such experimental evidence would require a direct measure of the interaction over various length scales (i.e. various $q$), which is challenging for electrons embedded in a solid. Short of that, our experiments suggest the presence of a minimum in $V(\mathbf{q})$, consistent with non-monotonic Coulomb interactions in the copper-oxide plane.

At a phenomenological level, we also note that the ring-like dynamic correlations bear a remarkable resemblance to a general theory considered by Brazovskii[45], where fluctuations of a finite discrete $\mathbf{q}$ order occur isotropically at $q = |\mathbf{q}|$. Examples range from magnetic order in MnSi[46] to the smectic order in liquid crystals[47], where the melting of the ordered state is due to temperature or magnetic field in the former, and due to electric field in the latter. For the CO correlations in Bi2212 the lattice plays the role of the external field, creating an additional potential that stabilizes the direction of static stripe domains. Furthermore, local defects may play a similar role, pinning dynamic circular correlations into short-range stripe domains[48].

The presence of Brazovskii fluctuations along all $\varphi$ may have direct implications to nematicity in the cuprates. In a configuration where most stripe domains align along $x$ rather than $y$ over a macroscopic-length scale, an effective nematic phase may occur[49,50]. Typically, one would consider this to be a competition between domains along only two directions, $x$ and $y$, but given the ability for the stripe fluctuations to occur along any $\varphi$, it should also be possible for nematic order to occur along with other directions. Thus, our findings may explain the multiple nematic directors observed across cuprate families and their interplay with charge order[51,52]. Overall, our observations suggest a mechanism where the combination of Coulomb interactions and lattice effects may lead to rotational symmetry breaking. This general mechanism could also play a role in the formation of nematic order in other strongly correlated systems, such as twisted bilayer graphene[53,54]. Beyond the interplay with nematic order, the existence of dynamic correlations over such a large circular manifold in momentum space should have other far-reaching consequences. For example, the intense three-dimensional CO peak in YBCO induced by external perturbations shows no evidence of spectral weight conservation[55–58]. Where does all the three-dimensional CO intensity come from? The ring-like intensity pattern provides a large manifold from which static CO correlations can emerge.

The quasi-circular in-plane electron–electron correlations in Bi2212 may reflect a general behavior of electrons in a polarizable medium (Fig. 4). This behavior could also be responsible for the emergence of various electronic phases in correlated materials, such as superconductivity or charge order. With special attention to the cuprates, our findings are also of immediate consequence to the interpretation of MEELS, EI-RXS, and RIXS experiments across several cuprate families. The effective Brazovskii structure of the electron fluctuations may also explain the puzzling observations of diagonal nematicity and three-dimensional CO. Finally, the competition between superconductivity and CO strongly suggests that the quasi-circular correlation are also important for superconductivity. Therefore, the ring-like dynamic correlations uncovered by our studies provide a centrally connected piece to the cuprate puzzle.

## Methods
RXS provides an enhanced sensitivity to charge modulations for a specific atomic species and orbital. The x-ray beam was tuned resonant to the Cu-$L_3$ edge ($\approx$932 eV) probing charge modulations within the $CuO_2$ plane[8].

**EI-RXS.** Energy-integrated Resonant x-ray scattering (EI-RXS) measurements were performed at the REIXS beamline of the Canadian Light Source (CLS) and at the ultrahigh vacuum diffractometer of the UE46-PGM1 beam line at the at BESSY II (Berliner Elektronenspeicherring für Synchrotronstrahlung) at the Helmholtz Zentrum Berlin, offering consistent results between different samples/end-stations. To maximize the CO diffraction signal, all measurements were performed in $\sigma$ geometry (photon polarization in the $a$–$b$ plane) and with the incoming photon energy tuned to the Cu-$L_3$ edge ($\approx$932 eV). The $\theta$ scans were performed with the detector angle fixed at 170°.

**RIXS.** RIXS measurements were performed at the qRIXS endstation at the ALS at the Lawrence Berkeley National Laboratory. The angle between beamline and spectrometer was set to 152°, which is the maximum value allowed by the instrument. The polarization was set along the scattering plane, $\pi$ geometry, due to limitations of the beamline. As in the EI-RXS measurements, different values of the in-plane momentum transfer were achieved by rotating the sample to different $\theta$ angles. At each $\theta$ the RIXS spectrum was measured for 90 s. This measurement was done for several values $\varphi$, the azimuthal angle, by rotating the sample about an axis perpendicular to its surface (i.e., the $c$ axis). Each $\theta$ scan required ~100 min for acquisition. Due to the instabilities of the beam, we measured the x-ray absorption spectrum prior to the acquisition of every $\theta$ scan to guarantee that the photon energy was constantly set to the maximum of the Cu-$L_3$ resonance. The elastic line position in the spectrometer was also independently confirmed by measurements on carbon tape placed next to the Bi2212 sample.

## Data availability
The datasets generated during and/or analysed during the current study are available from the corresponding author on reasonable request.

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

## Acknowledgements

We thank George Sawatzky, Rafael Fernandes, Steve Kivelson, Gergely Zimanyi, Rajiv Singh, Richard Scalettar, Riccardo Comin, Matteo Mitrano, and Michael Flynn for fruitful discussions. This research used resources of the Advanced Light Source, a DOE Office of Science User Facility under contract no. DE-AC02-05CH11231. The work at BNL is supported by US DOE, DE-SC0012704. The work at CRIEPI is supported by JSPS KAKENHI Grant Numbers JP17H01052. We thank HZB for the allocation of synchrotron radiation beamtime. Part of the research described in this paper was performed at the Canadian Light Source, a national research facility of the University of Saskatchewan, which is supported by the Canada Foundation for Innovation (CFI), the Natural Sciences and Engineering Research Council (NSERC), the National Research Council (NRC), the Canadian Institutes of Health Research (CIHR), the Government of Saskatchewan, and the University of Saskatchewan. This research was undertaken thanks in part to funding from the Max Planck-UBC-UTokyo Centre for Quantum Materials and the Canada First Research Excellence Fund, Quantum Materials and Future Technologies Program, in addition to the Killam, Alfred P. Sloan, and Natural Sciences and Engineering Research Council of Canada's (NSERC's) Steacie Memorial Fellowships (A. D.); the Alexander von Humboldt Fellowship (A.D.); the Canada Research Chairs Program (A.D.); NSERC, Canada Foundation for Innovation (CFI); British Columbia Knowledge Development Fund (BCKDF); and the Canadian Institute for Advanced Research (CIFAR) Quantum Materials Program. A.F. acknowledges support from the Alfred P. Sloan Fellowship in Physics and the UC San Diego startup funds. E.H.d.S.N. acknowledges prior support from the CIFAR Global Academy, the Max Planck-UBC postdoctoral fellowship, and UC Davis startup funds, as well as current support from the Alfred P. Sloan Fellowship in Physics.

## Author contributions

A.D., B.K., and E.H.d.s.N. conceived the EI-RXS experiments, which were performed by E.H.d.S.N., F.B., M.Min., R.S., E.S., M.B., and M.Mic with the help of E.W. and F.H. E.H. d.S.N. and A.F. designed the RIXS experiments, which were performed by E.H.d.S.N., S.D., and Y.Y. with the help of Y.C.S., X.F., and Y.D.C. F.B. performed and interpreted the theoretical calculations with the help of E.H.d.S.N. and A.F. The studied materials were synthesized by S.O., R.D.Z., J.A.S., and G.G. E.H.d.S.N., A.F., and F.B. wrote the manuscript with inputs from all the co-authors.

## Competing interests

The authors declare no competing interests.
