## [Peer Review File · Nature Communications]

REVIEWERS' COMMENTS

Reviewer #1 (Remarks to the Author):

The authors addressed all my remarks from the previous round of review and I think that their responses (also to the other reports) are factual and to the point.

I do not have other criticism and I recommend publication of the revised manuscript in its present form.

Reviewer #2 (Remarks to the Author):

I appreciate the authors' effort of making an extensive response to the referees' comments and substantially revising the manuscript. The authors admit now that they cannot unambiguously resolve the full momentum structure of $V(q)$. They make clear in the revised version of their manuscript that the ring-like charge order scattering response in their RIXS data is maybe a consequence of 'a specific feature' of the Coulomb potential, i.e. minima in $V(q)$. Now the authors also relativize their simulation as only a "proof-of-principle" calculation and clarify that from these data $V(q)$ cannot conclusively be extracted. The fact that the authors retract some of their earlier claims clearly demonstrates that their largely speculative conclusions are much less important than claimed. Despite the authors being able to alleviate most of my technical concerns, I still have serious concerns regarding a claimed substantial advance represented in this work compared to such CDW scattering patterns already previously observed in underdoped T^{\prime} -Nd₂CuO₄ thin films. I am still of the opinion that this study will likely only be of interest to a more specialized audience, but not to a more broad audience from several areas of physics. Despite finding the revised manuscript being much improved and attesting a useful message for the cuprate community to this manuscript, I am not at all convinced of the authors' arguments on the large advance and the high relevance of this work for a larger natural science community.

Reviewer #3 (Remarks to the Author):

Dear Editor,

I have read all the changes to the manuscript and now recommend publication. The experimental results, as I mentioned previously, are fascinating and the theoretical explanation is both simple but also now put in a suitable place as just a potential explanation of the data. The authors have addressed all my concerns about the original version of the manuscript.

I have only one minor superficial comment:

The first few sentences of the abstract should be improved for grammar and style. While I do not disagree with their content, they do not invoke confidence in the reader's mind and leave a good first impression. For example, the first line:

In strongly correlated systems the Coulomb interactions between electrons, relative to their kinetic energy, play a central role in determining their emergent quantum mechanical phases.

Should read

In strongly correlated systems, the ratio of the total Coulomb potential energy between electrons to their kinetic energy plays a central role in determining their emergent quantum mechanical phases.

or

In strongly correlated systems, the strength of the Coulomb interactions between electrons, relative to their kinetic energy, play a central role in determining their emergent quantum mechanical phases.

Also the phrase "exact knowledge" in: "requires the exact knowledge of the effective Coulomb interaction between electrons" is too high a standard.

Reply to Reviewers:

Reviewer comments are in black. Our responses are in *italic blue*.

We thank all the Reviewers for the insightful comments and suggestions, which helped us improve the manuscript.

Reviewer #1 (Remarks to the Author):

The authors addressed all my remarks from the previous round of review and I think that their responses (also to the other reports) are factual and to the point.

I do not have other criticism and I recommend publication of the revised manuscript in its present form.

We thank the Reviewer for all the previous comments and suggestions. We are happy to see the Reviewer now recommends the revised manuscript for publication.

Reviewer #2 (Remarks to the Author):

I appreciate the authors' effort of making an extensive response to the referees' comments and substantially revising the manuscript. The authors admit now that they cannot unambiguously resolve the full momentum structure of $V(q)$. They make clear in the revised version of their manuscript that the ring-like charge order scattering response in their RIXS data is maybe a consequence of 'a specific feature' of the Coulomb potential, i.e. minima in $V(q)$. Now the authors also relativize their simulation as only a "proof-of-principle" calculation and clarify that from these data $V(q)$ cannot conclusively be extracted. The fact that the authors retract some of their earlier claims clearly demonstrates that their largely speculative conclusions are much less important than claimed. Despite the authors being able to alleviate most of my technical concerns, I still have serious concerns regarding a claimed substantial advance represented in this work compared to such CDW scattering patterns already previously observed in underdoped T' -Nd₂CuO₄ thin films. I am still of the opinion that this study will likely only be of interest to a more specialized audience, but not to a more broad audience from several areas of physics. Despite finding the revised manuscript being much improved and attesting a useful message for the cuprate community to this manuscript, I am not at all convinced of the authors' arguments on the large advance and the high relevance of this work for a larger natural science community.

We thank the Reviewer for all the previous comments and suggestions. We are glad that the revised manuscript now alleviates the Reviewer's technical concerns. We are also happy to see that we were able to clarify our intended message in the revised manuscript and through our response to the Reviewer's report. We respectfully but firmly disagree with the Reviewer's assessment of the advance provided by our work. As detailed in the previous response to the Reviewer's comments, our work differs from the previous reports in electron-doped T' -NCO in significant ways, from the identification of the dynamic (inelastic) character of the "ring" scattering pattern to an alternate interpretation of the physical mechanism that originates the quasi-circular pattern (Coulomb interactions instead of a Fermi surface instability). Most importantly our observation in hole-doped cuprates actually suggests that the ring-like scattering effect may be universal and, therefore, even more important to the cuprate problem than originally thought. In the cuprates, it is important to understand what features are universal (e.g. high- T_c).

superconductivity) and what features are material-specific. In this sense, our work also adds to the importance of the T^ -NCO results.*

Reviewer #3 (Remarks to the Author):

Dear Editor,

I have read all the changes to the manuscript and now recommend publication. The experimental results, as I mentioned previously, are fascinating and the theoretical explanation is both simple but also now put in a suitable place as just a potential explanation of the data. The authors have addressed all my concerns about the original version of the manuscript.

I have only one minor superficial comment:

The first few sentences of the abstract should be improved for grammar and style. While I do not disagree with their content, they do not invoke confidence in the reader's mind and leave a good first impression. For example, the first line:

In strongly correlated systems the Coulomb interactions between electrons, relative to their kinetic energy, play a central role in determining their emergent quantum mechanical phases.

Should read

In strongly correlated systems, the ratio of the total Coulomb potential energy between electrons to their kinetic energy plays a central role in determining their emergent quantum mechanical phases.

or

In strongly correlated systems, the strength of the Coulomb interactions between electrons, relative to their kinetic energy, play a central role in determining their emergent quantum mechanical phases.

Also the phrase "exact knowledge" in: "requires the exact knowledge of the effective Coulomb interaction between electrons" is too high a standard.

We thank the Reviewer for all the previous comments and suggestions. We are happy to see the Reviewer recommends the revised manuscript for publication. We have also modified the first sentence of the abstract, following the Reviewer's suggestion. The new sentence is:

"In strongly correlated systems the strength of Coulomb interactions between electrons, relative to their kinetic energy, plays a central role in determining their emergent quantum mechanical phases."